# Lessons Learned from an Experience with Vancomycin-Intermediate *Staphylococcus aureus* Outbreak in a Newly Built Secondary Hospital in Korea

**DOI:** 10.3390/pathogens10050564

**Published:** 2021-05-06

**Authors:** Min Hyung Kim, Yong Chan Kim, Heejung Kim, Hyuk Min Lee, Ju Hyun Lee, Da Ae Kim, Chanhee Kim, Jin Young Park, Yoon Soo Park

**Affiliations:** 1Department of Internal Medicine, Division of Infectious Disease, Bundang Jesaeng Hospital, Seongam, Gyeonggi 13590, Korea; tormyday1@gmail.com; 2Department of Internal Medicine, Division of Infectious Disease, Yongin Severance Hospital, Yonsei University College of Medicine, Yongin-si 16995, Korea; AMOMJ@yuhs.ac; 3Center for Digital Health, Yongin Severance Hospital, Yonsei University Health System, Yongin-si 16995, Korea; EMPATHY@yuhs.ac; 4Department of Laboratory Medicine, Yongin Severance Hospital, Yonsei University College of Medicine, Yongin-si 16995, Korea; hjkim12@yuhs.ac; 5Department of Laboratory Medicine and Research Institute of Bacterial Resistance, Yonsei University College of Medicine, Seoul 03722, Korea; HMLEE71@yuhs.ac; 6Infection Control Office, Yongin Severance Hospital, Yongin-si 16995, Korea; MIDNIGHTDREAM@yuhs.ac (J.H.L.); VIEJTY@yuhs.ac (D.A.K.); 7Division of Disease Control Policy, Bureau of Health, Gyeonggi Provincial Office, Gyeonggi 13494, Korea; chanii.kim@gg.go.kr

**Keywords:** vancomycin-intermediate *Staphylococcus aureus*, healthcare-associated infection, outbreak, infection control, antimicrobial resistance

## Abstract

A vancomycin-intermediate *Staphylococcus aureus* (VISA) outbreak occurred in an intensive care unit (ICU) in South Korea. We aimed to investigate the condition that led to the VISA outbreak and seek measures to prevent further spread of the multidrug-resistant organism. A total of three VISA isolates were obtained from two patients and a health care worker (HCW) in a newly built 450-bed secondary hospital. Extensive screening of close contacts for VISA in terms of space sharing and physical contact, irrespective of contact time, was performed. Furthermore, multilocus sequence type, staphylococcal cassette chromosome mec type, and *spa* type profiles were determined for all VISA isolates. The relationship between vancomycin use and the minimum inhibitory concentration (MIC) of *S. aureus* was also investigated. Molecular typing showed that the strains of the three VISA isolates were identical, indicating horizontal hospital transmission. We assumed that VISA colonised in the HCW could have transmitted to the two patients, which resulted in one infection and one colonisation. The affected HCW was excused from work and was decolonised with mupirocin. Five weeks after the interventions, no additional VISA isolates were identified. No relationship between vancomycin use and MIC of *S. aureus* was identified. Extensive screening of contacts in addition to decolonisation is crucial in preventing the further spread of VISA.

## 1. Introduction

Antimicrobial resistance (AMR) is a global concern threatening public health. Infections caused by bacteria with AMR increase the risk of death and the costs of health care [1]. Methicillin-resistant *Staphylococcus aureus* (MRSA) is one of the major pathogens with AMR that infects humans. Vancomycin has been used as the first-choice drug to treat MRSA infection for decades. With the spread of MRSA infection worldwide, the empirical use of vancomycin has increased [2]. Consequently, the selective pressure of vancomycin has led to the emergence of *S. aureus* with reduced susceptibility to vancomycin, such as vancomycin-intermediate *S. aureus* (VISA) [3,4]. Following the first report of VISA from Japan in 1997, additional cases have been reported from several countries over the past two decades [5]. In South Korea, VISA was first isolated in 1998. Since it was designated as a notifiable bacterium with AMR in 2000, the number of reports has increased. Domestic cases of VISA have been sporadically reported across the country, but there have not been any previous reports of a VISA outbreak [6,7]. As a result, evidence regarding how to respond to VISA outbreaks is lacking [8,9]. Previous reports on VISA outbreaks suggested that considerable time and effort were required to control them due to the challenges in the early detection of the event and the high transmissibility of the pathogen.

Here, we describe the first outbreak of VISA that occurred in the intensive care unit (ICU) of a newly built secondary care hospital in South Korea. Compared with previous reports, this outbreak was rapidly controlled due to the rapid detection of colonising bacterial strains and prompt response. Based on the lessons learned from this experience, we aimed to investigate the conditions that lead to VISA outbreaks and to identify efficient measures to prevent further spread of the multidrug-resistant organism.

## 2. Results

### 2.1. Clinical Characteristics of Subjects with Vancomycin-Intermediate Staphylococcus aureus Isolates

During the study period, three VISA isolates were obtained from two patients and one health care worker (Figure 1). The clinical characteristics of the three individuals with VISA infection are described in Table 1. VISA was isolated from the sputum of a 66 year old man (Patient 1) with pneumonia. A subsequent nasal surveillance culture revealed that he was not an MRSA coloniser. Glycopeptides had previously been administered to the patient for 9 days for pneumonia treatment before the emergence of VISA. Although the vancomycin was switched with linezolid after the isolation of the VISA strain from his sputum, he died. Another VISA isolate was obtained from Patient 2, a 46 year old man; this was considered a transmitted case of colonisation without the signs of pulmonary infection. He died due to an intracerebral haemorrhage (his admission diagnosis). Among health care workers who were subjects of the contact investigation, the nasal cavity swab culture from a nurse (Nurse A) was positive for VISA. She was the nurse in charge of Patient 2, and participated in taking care of Patient 1. To eliminate the carriage of VISA, intranasal mupirocin was applied twice a day for 5 days. Repeat follow-up specimens were obtained over 3 weeks after initiating treatment, and all culture results were negative for VISA.

### 2.2. Microbiological Characteristics of Vancomycin-Intermediate Staphylococcus aureus

The bacteriological characteristics were evaluated to determine the antimicrobial resistance profiles and molecular types of the VISA isolates (Table 1). Antimicrobial susceptibility tests of VISA were performed for 17 antimicrobial agents. The minimum inhibitory concentration (MIC) of vancomycin was 4 μg/mL in all isolates. Two of the isolates were susceptible to teicoplanin, and the other had intermediate resistance to teicoplanin (MIC of 16 μg/mL). All isolates were susceptible to linezolid, quinupristin/dalfopristin, rifampicin, trimethoprim/sulfamethoxazole, and nitrofurantoin, and had high levels of resistance to ciprofloxacin, clindamycin, erythromycin, gentamicin, and oxacillin.

The multilocus sequence typing (MLST) patterns of the three isolates were identical (sequence type (ST) 5). Similarly, all three isolates were of the identical staphylococcal cassette chromosome mec (SCC*mec*) type (type II) and *spa* type (t2460).

### 2.3. Amount of Vancomycin Used and Changes in the Vancomycin Minimum Inhibitory Concentration for Staphylococcus aureus

The trends in vancomycin consumption and vancomycin MIC for *S. aureus* isolates are shown in Figure 2. The amount of vancomycin used increased from 49.15 defined daily dose (DDD) per 1000 patient days (PDs) in March 2020 to 78.52 DDD per 1000 PDs in June 2020. However, there was a marked decrease in vancomycin use in July and August 2020, to 45.60 and 36.87 DDD per 1000 PDs, respectively. A total of 160 *S. aureus* isolates were obtained from March 2020 to August 2020. The percentage of isolates with a vancomycin MIC of 1 μg/mL continuously increased from 65% to 96% and then decreased to 55%. The opposite trend was observed in the distribution of isolates with a vancomycin MIC of ≤0.5 μg/mL during the same period. Isolates with a vancomycin MIC of 2 μg/mL began to appear in July 2020. 

## 3. Discussion

This is the first study describing a VISA outbreak in an ICU in South Korea. We presume that the nurse colonised with VISA was responsible for VISA transmission to the two ICU patients. One patient developed a VISA infection, and the other patient was colonised by VISA. Prompt contact investigation and mupirocin decolonisation resulted in the early termination of the outbreak. About 5 weeks after the first detection of VISA, the outbreak ended and no further VISA isolates were reported during the 3 months of follow-up.

Previous studies have shown that most VISA strains emerged in patients with MRSA undergoing prolonged therapy with vancomycin [10,11]. However, the increased incidence of VISA is not necessarily attributable solely to increased vancomycin use. One epidemiological study found that nearly half of the VISA isolates were from patients without a history of MRSA infection or vancomycin use [6,12]. Consistent with previous reports, our study found that the amount of vancomycin used had no direct effect on the MIC of *S. aureus* isolates in the hospital. All isolates in our study were the same strains, which indicated horizontal transmission within the hospital. Horizontal transmission can be the cause of VISA emergence. As a newly built institution is not likely to have a reservoir for multidrug-resistant organisms, it is reasonable to assume that the VISA was introduced from an outside source through a person instead of acquired from the environment. This led us to conclude that the best prophylactic measure to prevent an outbreak of VISA is prompt screening with a nasal swab of the ICU patients and staff working in the ICU when VISA is first detected. Monitoring the usage of vancomycin or investigating the MRSA prevalence may not be an efficient means of preventing VISA emergence.

Previous studies of VISA outbreaks have highlighted the importance of isolation and strict contact precaution to terminate large-scale outbreaks [8,9]. One study described a VISA outbreak that affected 21 inpatients in a French hospital [8]. Another study, performed in a hospital in Japan, described a VISA outbreak in which 19 VISA isolates were obtained from 17 inpatients [9]. Although the staff involved in the control of the outbreak in the two hospitals implemented a variety of infection control measures, partial closure of the ICUs were unavoidable, and it took several months to eradicate the outbreaks. Compared with our study, the first case of VISA was recognised relatively late, which may have contributed to the long time taken to end the outbreak. For early detection of VISA cases, extensive investigation screening for colonisers on detection of the first case is imperative not only in contacts who have spent extended time with a case, but also in those who have shared fomites with a case.

The molecular epidemiology and resistance profiles of VISA in South were described in a previous study [13]. The most prevalent molecular type was ST5-SCC*mec* type II-t2460, followed by ST72-SCC*mec* type IV-t324. A recent study, which used national data of VISA strains isolated between 2014 and 2016, reported similar results [6]. Concordant with the national molecular epidemiology, all VISA isolates in our study were of the ST5-SCC*mec* type II-t2460 genotype, which may have originated in a healthcare setting [14]. As for the resistance profiles of VISA, all the VISA strains were multidrug resistant in our study. However, consistent with previous studies [6,13], we found that all three VISA isolates were susceptible to linezolid. This finding indicates that linezolid can be used as a therapeutic option for VISA infections.

For asymptomatic individuals colonised with *S. aureus*, nasal decolonisation with mupirocin is a potential strategy that may be used to prevent infection [15,16]. Intranasal mupirocin can lead to a reduction in the nasal carriage of *S. aureus*. However, limited information is available regarding the effectiveness of mupirocin for VISA decolonisation. In our study, a healthcare worker colonised with VISA (mupirocin MIC 4 μg/mL) was treated with mupirocin; subsequently, three cultures for VISA were negative over a 3 week period. Furthermore, no additional VISA isolates were identified in the 3 months after her return to the hospital. The effectiveness of intranasal mupirocin for VISA decolonisation needs to be determined by means of further studies.

We demonstrated an association between the amount of vancomycin consumption and the vancomycin MIC of *S. aureus*. Vancomycin MIC creep was observed among S. *aureus* isolates in our hospital during the period of increased vancomycin use. Similar findings have been reported in other studies [17,18]. However, our results should be interpreted with caution. The observation period was too short to conclude that there was a relationship between the level of vancomycin consumption and the vancomycin MIC of *S. aureus*. In addition, the VISA outbreak strain emerged in the presence of decreasing vancomycin consumption. Further observation over a longer time period is necessary to evaluate the relationship between the amount of vancomycin use and the VISA isolation rate.

There were some limitations in this study. First, both patients died early, obscuring the actual effect of enhanced infection control measures. Second, the small number of cases prevented us from further investigating the mode of VISA transmission. Third, we were unable to perform genetic analysis for specific genes that could have determined the mechanism involved in antimicrobial resistance. However, our study has significance in that it showed the importance of early detection of VISA in preventing a large outbreak.

The isolation of VISA strains has been continuously reported in South Korea since 1998. This report describes the first VISA outbreak in South Korea, which occurred through horizontal transmission in the ICU of a newly built secondary care hospital. Early detection of the outbreak through an extensive search for colonisers through nasal swab screening followed by the implementation of effective infection control measures (including mupirocin decolonisation) resulted in the successful control of the outbreak. Further studies on the effectiveness of mupirocin application in preventing the spread of VISA are required. Clinicians should suspect a VISA outbreak if multiple cases of VISA infection are identified in a healthcare setting within a short period of time.

## 4. Materials and Methods

### 4.1. Setting

This study was conducted at Yong-in Severance Hospital, South Korea, which was opened in March 2020. The hospital has 450 beds, including an ICU with 19 beds that was in operation at the time of outbreak. The ICU consists of 15 large private rooms and four open beds separated by curtains. Critically ill patients were admitted to the ICU because of their medical or surgical conditions. There were 3 critical-care specialists, 24 nurses, and 5 nurse assistants working in the ICU.

### 4.2. Outbreak and Intervention

The first patient (Patient 1) identified with VISA was admitted to the hospital on 12 August 2020 for acute myeloid leukaemia and was referred to the ICU on day 5 due to pneumonia requiring mechanical ventilation. A strain of VISA was detected from the patient’s sputum 9 days after ICU admission. On the following day, an isolate of VISA was reported from the sputum of another patient (Patient 2) who had been in the ICU for 17 days and was on mechanical ventilation. Transmission was suspected, as the two patients were staying in adjacent private rooms in the ICU (Figure 3). Infection control interventions were implemented immediately after detection of the second VISA isolate.

A contact investigation was performed soon after report of the second case. Contacts were classified into three categories based on the level of interaction with the VISA patients using the Real-Time Location System tracing system: extensive, moderate, and minimal [19]. We conducted the investigation in a step-wise manner. Nasal swab specimens were collected sequentially, starting with those who had had extensive contact with the patients from whom VISA had been isolated. If positive results were identified among these contacts, we extended the scope of the investigation to the next category. Among 24 contacts, one member of the nursing staff was classified as having extensive contact with Patient 2 but minimal contact with Patient 1, and was found to be positive for VISA on nasal swab culture. Therefore, specimen collection was extended to 23 persons who had had moderate interactions with the patients of interest. All samples collected from 23 persons tested negative for VISA. In hindsight, we concluded that the member of the nursing staff with the positive culture was probably the source case of the VISA outbreak. However, no further isolates of VISA were cultured among patients who the nurse had taken care of or among her co-workers. Environmental samples were also collected from surfaces and equipment in the VISA patients’ rooms as well as from work spaces in the ICU (bed fence, infusion pump, medication carts, dressing trolleys, telephones, monitor panels, automatic door buttons, computer keyboards, and electrocardiography machines).

The nurse with a positive VISA result was excused from patient-care activities. Contact precautions were implemented for the patients from whom VISA was isolated. All health care workers were required to wear gowns and gloves before contacting these patients. Hand hygiene was reinforced and monitored by an infection control team. The ICU staff were provided education on infection control policies for VISA.

### 4.3. Laboratory Tests for Staphylococcus aureus

The MIC of an antimicrobial agent was determined in the hospital using a VITEK-2 analyser (bioMérieux, Hazelwood, MO, USA). Antimicrobial susceptibility testing for *S. aureus* was performed for penicillin, oxacillin, gentamicin, ciprofloxacin, erythromycin, telithromycin, clindamycin, linezolid, teicoplanin, vancomycin, tetracycline, tigecycline, nitrofurantoin, mupirocin, rifampicin, quinupristin/dalfopristin, and trimethoprim/sulfamethoxazole. The susceptibility results were interpreted according to the Clinical and Laboratory Standards Institute guidelines [20]. All *S. aureus* strains with a vancomycin MIC of ≥4 μg/mL were sent to the Institute of Health and Environment for confirmatory testing using broth microdilution. *S. aureus* strains with a vancomycin MIC of 4–8 μg/mL, confirmed by the broth microdilution method, were defined as VISA [21].

Genomic deoxyribonucleic acid (DNA) was extracted using a Wizard genomic DNA preparation kit (Promega, Madison, WI, USA). According to the manufacturer’s protocol for bacterial cells, we added lysostaphin at the final concentration of 30 μg/mL in the lysis buffer and incubated at 37 °C for 1 h. MLST, SCC*mec* typing, and *spa* typing were performed in the present study. MLST was carried out by polymerase chain reaction (PCR) amplification and sequencing of seven housekeeping genes (*arc*, *aroE*, *glpF*, *gmk*, *pta*, *tpi*, and *yqiL*) using primer pairs as previously described [22]. The allelic profiles and STs were assigned by the MLST website (http://saureus.mlst.net/, accessed on 26 October 2020). SCC*mec* types were determined by the multiplex PCR method [23]. Strains COL, N315, NCCP13860, and MW2 were included as controls for SCC*mec* types I, II, III, and IV, respectively. The *spa* typing was performed as previously described [24,25]. The *spa* types were determined using Ridom SpaServer (http://spa.ridom.de/spatypes.shtml, accessed on 26 October 2020).

### 4.4. Vancomycin Consumption and the Vancomycin Minimum Inhibitory Concentrations of Staphylococcus aureus Isolates

We investigated the amount of vancomycin use and the changes in vancomycin MICs in *S. aureus* between March 2020 and August 2020. Data were collected on the monthly prescription of vancomycin to evaluate the trend in vancomycin use. The DDD of vancomycin was determined according to the World Health Organization Anatomical Therapeutic Chemical Classification [26]. The amount of vancomycin consumption is expressed as the monthly DDD per 1000 PDs. We collected data for vancomycin MICs of *S. aureus* isolated from clinical specimens among patients in the hospital. All patients with *S. aureus* infection were included in the analysis only once. For patients with isolates identified repeatedly, only the first isolate was tested.

### 4.5. Case Definition

The presence of infection with VISA was determined according to the Centers for Disease Control and Prevention/National Healthcare Safety Network surveillance criteria [27]. Colonisation was defined as the isolation of VISA from clinical samples without interaction between the host and organism, i.e., in the absence of clinical symptoms or an immune response.

### 4.6. Ethics

The study protocol was approved by the Institutional Review Board of Yonsei University Health System Clinical Trial Centre (approval number: 9-2020-0141, approved on 15 December 2020). As this study was retrospective and the study participants were anonymised, the Institutional Review Board waived the requirement for patient consent.

## Figures and Tables

**Figure 1 pathogens-10-00564-f001:**
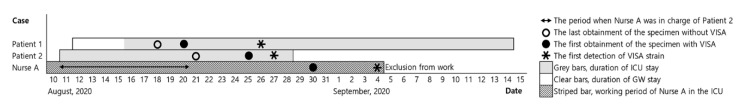
Time course of the vancomycin-intermediate *Staphylococcus aureus* outbreak: GW, general ward; ICU, intensive care unit; VISA, vancomycin-intermediate *Staphylococcus aureus*.

**Figure 2 pathogens-10-00564-f002:**
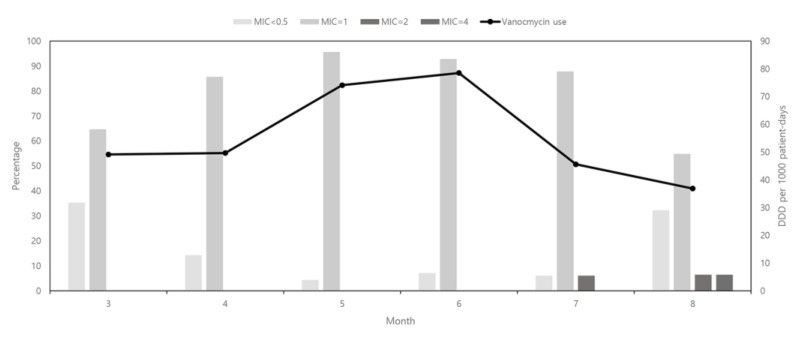
The relationship between the amount of vancomycin used and the vancomycin MIC of *Staphylococcus aureus* isolates. MIC, minimum inhibitory concentration.

**Figure 3 pathogens-10-00564-f003:**
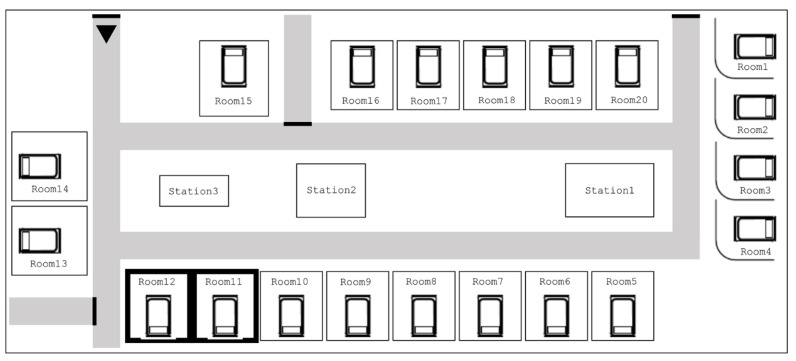
Room arrangement in the intensive care unit. The first vancomycin-intermediate *Staphylococcus aureus* isolate was obtained from a patient in room 11, and the second isolate was obtained from a patient in room 12.

**Table 1 pathogens-10-00564-t001:** Clinical and microbiological features of the vancomycin-intermediate *Staphylococcus aureus* cases identified during the outbreak.

Variable	Patient 1	Patient 2	Nurse A
Clinical characteristics			
Age (years)	66	46	34
Sex	Male	Male	Female
Specimen	Sputum	Sputum	Nasal cavity
Diagnosis on ICU admission	Acute myeloid leukaemia, pneumonia	Intracerebral haemorrhage, hypertension	None
Risk factors associated with VISA	Previous vancomycin use, indwelling medical devices	Indwelling medical devices	None
Case definition	Presence of infection	Colonisation	Colonisation
Outcome	Died	Died	Decolonisation
Antimicrobial resistance profiles of VISA			
Ciprofloxacin	≥8	≥8	≥8
Clindamycin	≥8	≥8	≥8
Erythromycin	≥8	≥8	≥8
Telithromycin	≥4	≥4	≥4
Gentamicin	≥16	≥16	≥16
Mupirocin	4	4	4
Oxacillin	≥4	≥4	≥4
Penicillin G	≥0.5	≥0.5	≥0.5
Quinupristin/dalfopristin	0.5	0.5	≤0.25
Rifampicin	≤0.5	≤0.5	≤0.5
Trimethoprim/Sulfamethoxazole	≤10	≤10	≤10
Tetracycline	≥16	≥16	≥16
Tigecycline	1	0.5	0.5
Nitrofurantoin	32	32	≤16
Teicoplanin	16	8	8
Vancomycin	4	4	4
Linezolid	4	4	2
Molecular features of VISA			
MLST	ST5	ST5	ST5
SCC*mec* type	II	II	II
*spa* type	t2460	t2460	t2460

ICU, intensive care unit; MLST, multilocus sequence typing; SCC*mec*, Staphylococcal cassette chromosome mec; VISA, vancomycin-intermediate *Staphylococcus aureus*.

## Data Availability

Data presented in this study are available on request from the corresponding author.

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
