# Peer review of "Lessons Learned from an Experience with Vancomycin-Intermediate Staphylococcus aureus Outbreak in a Newly Built Secondary Hospital in Korea"

_pathogens, 2021, doi:10.3390/pathogens10050564_

Round 1
Reviewer 1 Report
Were Human ethics obtained for this study?
The VISA strains usually have specific genes, it would be good to do some simple studies to clearly identify these are present and the DNA nature of these.
I am interested in the basis (or reasoning) for the screening of the patients/HCW. Was a large screening undertaken across the hospital and ICU?
Further to this. The nurse who treated patient 1 and patient 2 would have also had other patients - where they also screening. Were all the subjects screening in the nasal passage and skin?
While I understand the data and discussion around Figure 2 it is poorly presented and needs to be re-done.
Author Response
Reviewer #1:
- Were Human ethics obtained for this study?
Response: This study was approved by the Institutional Review Board of Yonsei University Health System Clinical Trial Centre. We added this information in our manuscript as follows.
In the “Ethics” subsection of the Material and Methods section:
“4.6. Ethics
The study protocol was approved by the Institutional Review Board of Yonsei University Health System Clinical Trial Centre (approval number: 9-2020-0141). As this study was retrospective and the study participants were anonymised, the Institutional Review Board waived the requirement for patient consent. “
- The VISA strains usually have specific genes, it would be good to do some simple studies to clearly identify these are present and the DNA nature of these.
Response: We highly appreciate your recommendation. Due to time and resource constraints, we omitted this analysis which might have made a useful addition to our research. We mentioned this limitation in the Discussion section.
In the Discussion section:
“Third, we were unable to perform genetic analysis for specific genes that could have determined the mechanism involved in antimicrobial resistance. “
- I am interested in the basis (or reasoning) for the screening of the patients/HCW. Was a large screening undertaken across the hospital and ICU?
Response: We conducted the investigation in a step-wise manner. Contacts were classified into three categories (extensive, moderate, and minimal) based on their level of interaction with the VISA patients, then screening nasal swabs was performed sequentially, starting with people in the extensive contact category. If positive results were identified among these contacts, we extended the scope of the investigation to the next category. We elaborated in manuscript as follows.
On ‘Outbreak and Intervention’ section in Material and Methods
“We conducted the investigation in a step-wise manner. Nasal swab specimens were collected sequentially, starting with those who had had extensive contact with the patients from whom VISA had been isolated. If positive results were identified among these contacts, we extended the scope of the investigation to the next category.”
- Further to this. The nurse who treated patient 1 and patient 2 would have also had other patients - where they also screening. Were all the subjects screening in the nasal passage and skin?
Response: We screened all 27 patients and 23 co-workers who fell in the extensive contact category. The timeline of contact tracing started from the date when the first VISA patients was admitted to the ICU with the intention of extended the investigation upon the identification of positive results. We mentioned the result in the Material and Methods section as follows:.
In the “Outbreak and investigation” subsection of the Material and Methods section
“However, no further isolates of VISA were cultured among patients who the nurse had taken care of or among her co-workers.”
- While I understand the data and discussion around Figure 2 it is poorly presented and needs to be re-done.
Response: We appreciate your comment. We revised the manuscript as follows:
In the Discussion section:
“Vancomycin MIC creep was observed among S. aureus isolates in our hospital during the period of increased vancomycin use. Similar findings have been reported in other studies. However, our results should be interpreted with caution. The observation period was too short to conclude that there was a relationship between the level of vancomycin consumption and vancomycin MIC of S. aureus. In addition, the VISA outbreak strain emerged in the presence of decreasing vancomycin consumption. Further observation over a longer time period is necessary to evaluate the relationship between the amount of vancomycin use and VISA isolation rate.”
Reviewer 2 Report
I think it's a clear example of a great case study of a recent VISA outbreak. Although VISA strains have been around since the 90s, there is not much data on outbreaks as they are relatively uncommon. This is a clear example how to take appropriate measures to deal with those outbreaks. I have no concerns on the methodology used in this manuscript
Author Response
Reviewer #2:
I think it's a clear example of a great case study of a recent VISA outbreak. Although VISA strains have been around since the 90s, there is not much data on outbreaks as they are relatively uncommon. This is a clear example how to take appropriate measures to deal with those outbreaks. I have no concerns on the methodology used in this manuscript
Response: Thank you for your detailed comments.